# Numerical Solution of Nonlinear Diff. Equations for Heat Transfer in Micropolar Fluids over a Stretching Domain

**Farooq Ahmad** [1,2,*], **A. Othman Almatroud** [1], **Sajjad Hussain** [2], **Shan E. Farooq** [3] **and Roman Ullah** [4]

1   Department of Mathematics, Faculty of Science, University of Ha'il, Ha'il 55425, Saudi Arabia; o.almatroud@uoh.edu.sa
2   School of Mechanical and Aerospace Engineering, NANYANG Technological University, Singapore 639897, Singapore; hussain.sajjad@ntu.edu.sg
3   Mathematics Department, Govt. College University, Lahore 54000, Punjab, Pakistan; shanefarooq@gmail.com
4   Department of Computing, Muscat College, Muscat 113, Oman; romanullah@yahoo.com
*   Correspondence: ahmad.farooq@uoh.edu.sa or ahmad.farooq@ntu.edu.sg

**Abstract:** A numerical study based on finite difference approximation is attempted to analyze the bulk flow, micro spin flow and heat transfer phenomenon for micropolar fluids dynamics through Darcy porous medium. The fluid flow mechanism is considered over a moving permeable sheet. The heat transfer is associated with two different sets of boundary conditions, the isothermal wall and isoflux boundary. On the basis of porosity of medium, similarity functions are utilized to avail a set of ordinary differential equations. The non-linear coupled ODE's have been solved with a very stable and reliable numerical scheme that involves Simpson's Rule and Successive over Relaxation method. The accuracy of the results is improved by making iterations on three different grid sizes and higher order accuracy in the results is achieved by Richardson extrapolation. This study provides realistic and differentiated results with due considerations of micropolar fluid theory. The micropolar material parameters demonstrated reduction in the bulk fluid speed, thermal distribution and skin friction coefficient but increase in local heat transfer rate and couple stress. The spin behavior of microstructures is also exhibited through microrotation vector $N(\eta)$.

**Keywords:** solution of nonlinear equations; micropolar fluids; similarity transformations; porous medium; heat transfer; permeable stretching sheet

## 1. Introduction

The scientific and technological advances have brought a great awakening of interest in constructing different types of fluids and investigating their flow behavior in various useful geometries. Fluids with microstructures behave differently from the classical fluids. The flow and heat transfer behavior of these fluids cannot be described adequately with the classical theory of Newtonian fluid flows. Several theories have been presented to describe the very nature of these fluids. However, theory of micropolar fluids presented by Eringen [1] provides ample details required for justification of dynamics of such fluids. Micropolar fluids consist of rigid, randomly oriented, spherical particles with their own spins and microrotation, suspended in a viscous medium. Here the microelements are allowed to undergo rigid rotations only without stretch. The micropolar fluid model, apart from usual

velocity vector involves a microrotation vector and a gyration parameter to simulate the kinematics of microrotation. These fluids possess monosymmetric stress tensor. Later, Eringen [2] extended his theory for thermo-micropolar fluids and derived the constitutive laws. Ariman et al. [3,4] presented an excellent review of micropolar fluids and their applications. Ahmadi [5] investigated the boundary layer flow of micropolar fluids past a semi-infinite plate. The basic theory of micropolar fluids can be viewed in the book written by Eringen [6] as well as by Be'g et al. [7]. Rehman et al. [8] considered heat transfer in two-dimensional steady hydromagnetic natural convection flow of a micropolar fluid past a non-linear stretching sheet with temperature dependent viscosity.

A porous medium is usually composed of a solid matrix and voids, it has relevance to heat and mass transfer applications in thermal and geophysical processes. The viscous flow through porous medium is described by the well-known Darcy law which was supported experimentally, numerically and theoretically [9,10] but as stated above, it is valid only for flows with a sufficiently small velocity or $R$ Reynolds number. The Darcy law has been formulated $\nabla p = -\frac{\mu}{K}u$ by Bear [11]. The constant in Darcy's equation was proved later by Muskat and Wyckoff [12] that is related to the permeability of the porous material. Liu [13] presented closed form solution for the flow and heat transfer of a viscous fluid saturated in a porous medium past a permeable and non-isothermal stretching sheet with internal heat generation or absorption and radiation. Mohamed and Kasseb [14] obtained numerical solution steady flow and heat transfer in a porous medium saturated with a Sisko nano fluid (non-Newtonian power-law) over a nonlinearly stretching sheet in the presence of heat generation/absorption. Ferdows et al. [15] investigated effects of thermal radiation on a steady boundary layer flow with temperature-dependent thermal conductivity due to a stretching sheet through porous medium the presence of transverse magnetic field near a stagnation point. Berre et al. [16] described detailed reviews on modeling approaches for flow in fractured porous media, from physical, conceptual and mathematical models with two discretization approaches. Sun [17] presented an extension of the explicit moving particle semi-implicit (MPS) method with model formulation, based on the local volume averaging equations to compute the macroscopic behaviors of incompressible flows in porous media.

Heat transfer is important in many industrial processes. Upendar and Srinivasacharya [18] analyzed a mathematical model for the steady, mixed convection heat and mass transfer along a semi-infinite vertical plate embedded in a micropolar fluid in the presence of a first-order chemical reaction and radiation. MHD flow of micropolar fluid past a stretching sheet with heat transfer and with suction/blowing through a porous medium has been studied by Aldabe et al. [19]. Sharma et al. [20] studied the fully developed electrically conducting micropolar fluid flow and heat transfer along a semi-infinite vertical porous moving plate including the effect of viscous heating and in the presence of a magnetic field applied transversely to the direction of the flow. Mohammedein and Gorla [21] analyzed the flow of micropolar fluids bounded by a stretching sheet with a prescribed wall heat flux, viscous dissipation and internal heat generation. Abo-Eldahab and El-Aziz [22] considered heat transfer effect in a micropolar fluid flow induced by a stretching surface immersed in a porous medium with uniform free stream. Ahmad et al. [23] obtained closed form solution for a viscous, incompressible, MHD flow over a porous stretching sheet. Mahapatra and Gupta [21] investigated the flow and heat transfer characteristics over a stretching sheet with a uniform magnetic field and prescribed surface heat flux. Dayyan et al. [24] studied the Newtonian fluid flow with heat transfer through porous medium and presented analytical solution by employing the Homotopy Analysis Method (HAM).

The mathematical formulation of fluid flow problems is based on the fundamental laws of conservation and the nonlinear characteristic is inherited in the governing equations. Thus, in general, the exact solution for the flow problems is difficult to obtain, as opined by Ren [25]. However, different numerical and analytical approximate methods have been adopted to analyze the nonlinearity in the flow process. Some most often used approximate analytical methods for analysis of flow problems include VIM (variational iteration method), HAM (homotopy analysis method), DTM (differential transformation method), ADM (Adomain decomposition method), VPM (variation of parameter method), OHAM (optimal homotopy asymptotic method), etc. However, the computational cost

and time of these methods is increased for the determination of the unknowns (included to meet the second boundary conditions). Runge Kutta (R-K) methods with or without shooting techniques have been widely utilized for the solution of flow problems [26–28]. However, the Runge–Kutta method is widely used to solve the ordinary differential equation [29]. Similarly, other basic numerical methods have their own limitations. In the meantime, the advent of powerful computers facilitated the implementation of various numerical approaches to handle the complexities in this research area. In this scenario, numeric computation is capable of yielding reliable and cost-effective numerical solutions. Here, the governing differential equations are discretized. The basic discretization methods are FDM (finite difference method), FVM (finite volume method), and FEM (finite element method). FDM is the oldest of the three techniques, it is easy to implement and works well for the simple grids. In FDM, the continuous derivative is discretized to represent the difference of the flow variable between neighboring cells [30]. Among others, Ahmad et al. [31] used a finite-difference scheme known as the Keller-box method to study the flow and heat transfer of a micropolar fluid past a nonlinearly stretching plate. Finite difference solution for micropolar flow problems are analyzed by Hussain et al. [32] and Ashraf et al. [33], Shafique and Rashid [34].

Motivated from the above cited studies, we revisited [24] for the non-linear analysis of thermal transportation in micropolar fluids' motion over a permeable stretching sheet related to porous media with wall temperature and heat flux boundary conditions. A finite difference based scheme which comprises successive over relaxation iterative procedure, Simpson's (1/3) rule and Richardson extrapolation is harnessed to yield the solution. The optimum value of the relaxation parameter $\omega_{opt}$ is estimated to accelerate the convergence of the SOR method. In computational terms, this scheme is cost effective and well established. It can be utilized for reliable solution of simultaneous non-linear equations. We obtained results for the velocity, skin friction coefficient temperature, local Nusselt number, microrotation and couple stress under the variation of controlling parameters. Section 2 presents mathematical analysis, Section 3 provides the detail of numerical method. Section 4 contains results with discussion. Accuracy for this solution is assured through their computations for three grids sizes ($h$, $h/2$, $h/4$) and their closeness to other published studies in the limiting case. In addition, the extrapolation routine raises the order of accuracy to higher level.

## 2. Mathematical Analysis

In the mathematical theory of micropolar fluids, there are, in general, six degrees of freedom, three for translation and three for microrotation. The essence of the theory of micropolarfluid flow lies in the extension of the constitutive equations for Newtonian fluids so that more complex with microstructure such as animal blood, muddy water, colloidal fluids, lubricants and chemical suspensions. The micropolar fluids consist of randomly oriented particles suspended in a viscous medium. In practice, the theory of micropolar fluids requires that one must add a transport equation representing the principle of conservation of local angular momentum to the usual transport equations for the conservation of mass and momentum, and additional local constitutive parameters are also introduced. In this way, it enables to recover the inadequacy of Navier–Stokes theory to describe the correct behavior of fluids with microstructures. The governing equations of the motion are [1]:

$$\partial \rho / \partial t + (\nabla . \rho \mathbf{V}) = 0 \text{ (Mass)} \tag{1}$$

$$(\lambda_1 + 2\mu + \kappa)\nabla(\nabla . \mathbf{V}) + \kappa \nabla \times \boldsymbol{\Omega} - \nabla p - (\mu + \kappa)\left[\nabla \times \nabla \times \mathbf{V}\right] + \rho\,\mathbf{f} = \rho\frac{D\mathbf{V}}{Dt} \text{ (Momentum)} \tag{2}$$

$$(\alpha + \beta + \gamma)\nabla(\nabla . \boldsymbol{\Omega}) - \gamma[\nabla \times \nabla \times \boldsymbol{\Omega}] + \kappa \nabla \times \mathbf{V} - 2\kappa\boldsymbol{\Omega} + \rho\mathbf{l} = \rho j\frac{D\boldsymbol{\Omega}}{Dt} \text{ (Local angular momentum)} \tag{3}$$

where $\mathbf{V}$ is velocity, $\boldsymbol{\Omega}$ is microrotation vectors, $\rho$ is the density, $\mathbf{f}$ is body force, $\mathbf{l}$ is body couple per unit mass, respectively, $p$ is pressure, $j$ is the micro-inertia, $\mu$, $\kappa$, $\lambda_1$, $\gamma$ are, respectively, dynamic viscosity, vortex viscosity, Stokes viscosity and the spin gradient viscosity. $\alpha$, $\beta$, $\lambda$ symbolize material constants.

Consider micropolar fluid flow through a homogeneous porous medium of permeability of $K$, over a porous stretching sheet. The flow is time independent, incompressible and two-dimensional. Fluid flows due to a permeable sheet of length $L$ which is stretching linearly along the $x-axis$. The $y-axis$ is perpendicular to the sheet. The linear velocity distribution of flow along the sheet is $u_w = u_0x/L$. The body force and body couple is neglected. The schematic diagram is shown in Figure 1.

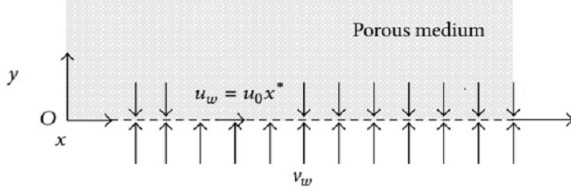

**Figure 1.** Schematic Diagram.

The Equations (1)–(3) are respectively simplified in view of the above assumptions to yield as below:

$$\partial u/\partial x + \partial v/\partial y = 0. \tag{4}$$

$$(\mu + k)(\partial^2 u/\partial x^2 + \partial^2 u/\partial y^2) + k\partial w_3/\partial y - \frac{\mu}{K}u = \rho(u\partial u/\partial x + v\partial u/\partial y). \tag{5}$$

$$\gamma\, \partial^2 w_3/\partial y^2 - 2kw_3 + k(\partial u/\partial x) = \rho j\, v\, \partial w_3/\partial y. \tag{6}$$

where $u$, $v$ are velocity components along horizontal and vertical directions and $\mathbf{\Omega} = \Omega(0,\ 0,\ \omega_3)$ is micro rotation vector perpendicular to $xy-$ plane, having resemblance to $\nabla \times \mathbf{V}$. The fluid temperature is $T$.

The energy equation:

$$u\partial T/\partial x + v\partial T/\partial y = (\alpha_{eff})\partial^2 T/\partial y^2. \tag{7}$$

The hydrodynamic boundary conditions are:

$$\left.\begin{array}{c} u(x^*,\ 0) = u_0x^*, \quad v(x^*,\ 0) = v_w, \quad \omega_3(x^*,\ 0) = -m\partial u/\partial y, \\ u(x^*,\infty) = 0, -\omega_3(x^*,\infty) = 0 \end{array}\right\} \tag{8}$$

where $u_0$ is constant, $v_w$ is mass transfer speed at the wall and $m$ is constant in the range $0 \le m \le 1$. The case $m = 0$ is called strong concentration by Guram and Smith [35], and indicates $N = 0$ near the surface and represents concentrated particle flows in which the microelements close to the surface are unable to rotate (Jena and Mathur [36]). The case $m = 1/2$ indicates the vanishing of antisymmetrical part of the stress tensor and denotes weak concentration (Ahmadi [5]). The case $m = 1$, as suggested by Peddieson [37], is used for the modeling of turbulent boundary-layer flows.

The thermal boundary conditions are:

$$\left.\begin{array}{ll} T(x^*,\ 0) = T_\infty + T_0(x^*)^s, & T(x^*,\infty) = T_\infty \quad \text{Power-law temperature} \\ -\tau\, \partial T/\partial y\big|_{(x^*,\ 0)} = q_0(x^*)^s, & T(x^*,\infty) = T_\infty \quad \text{Power-law heat flux} \end{array}\right\} \tag{9}$$

where $s$ is power law index and Tau ($\tau$) is the effective thermal conductivity of the medium and is a function of thermal conductivities of the fluid and solid phases and the porous medium microstructure. $x^* = x/L$. Let us take the similarity transformations as:

$$\psi = u_0x^* \sqrt{K}f(\eta) \tag{10}$$

where $\partial \psi / \partial y = u$, $-\partial \psi / \partial x = v$ and $\eta = \frac{y}{\sqrt{K}}$ is dimensionless variable.

$$\left.\begin{aligned} u &= u_0 x^* f'(\eta), \\ v &= \frac{u_0}{L} \sqrt{K} f(\eta), \\ w_3 &= \frac{u_0 x^*}{\sqrt{K}} N(\eta). \end{aligned}\right\} \tag{11}$$

The Equation (4) is readily satisfied. The Equations (5)–(7) are respectively transformed to ordinary differential forms:

$$(1 + C_1)\, f''' + R(f f'' - f'^2) - f' + C_1 N' = 0 \tag{12}$$

$$N'' - C_1 C_2 (f'' - 2N) = C_3 (f' N - f N') \tag{13}$$

$$\theta'' + (f\, \theta' - s f\,' \theta) P_r R = 0 \tag{14}$$

where prime denotes the differentiation with respect to $\eta$. $R = \frac{\rho u_0 K}{L\mu}$ is the Reynolds number, $C_1 = \frac{\kappa}{\mu}$, $C_2 = \frac{\kappa \mu}{\gamma}$ and $C_3 = \frac{\rho j \kappa \mu}{\gamma}$ are non-dimensional material constants.

The boundary conditions (8) and (9) then become:

$$\left.\begin{aligned} f(0) = \lambda, \ f'(0) = 1, \ N(0) = 0, \ f'(\infty) = 0 \ \text{and} \ N(\infty) = 0, \\ \theta(0) = 1, \theta(\infty) = 0, \qquad \text{Isothermal} \\ \theta'(0) = -1, \theta'(\infty) = 0, \quad \text{Isoflux} \end{aligned}\right\} \tag{15}$$

where $\lambda = -\frac{v_{wL}}{u_0 \sqrt{K}}$ is suction/injection parameter. Furthermore, $\lambda > 0$ shows suction and $\lambda < 0$ is for injection.

The physical quantities of significance or local wall shear stress $\tau_w$, the wall couple stress $m_w$, and the heat transfer from sheet surface $q_w$ as reported in [33].

$$\tau_w = -[(\mu + \kappa) \partial u / \partial y + \kappa N]_{y=0}, \ m_w = \gamma [\partial N / \partial y]_{y=0},$$
$$q_w = -[\tau \partial T / \partial y]_{y=0.}$$

## 3. Existence of Results

In order to establish the existence of a solution, the Equation (12) is formulated to the corresponding initial value problem (IVP), and the related boundary conditions in (15) are rewritten as:

$$f(0) = \lambda, \ f'(0) = 1, -f''(0) = \delta, \ N(0) = 0 \tag{16}$$

The parameter $\delta$ corresponds to coefficient of skin friction. This IVP formulation enables us to utilize topological shooting argument [38]. The solution of Equation (12) along with (16) is denoted by $f(\eta, \delta)$.

Then it is to show that a choice of $\delta$ can be made such that $f'(\eta, \delta)$ exists, for all $\eta > 0$ and $f'(\infty) = 0$ is satisfied. To achieve this, two sets are defined:

$$X = \{\delta < 0 / \text{there is first point } \eta_X > 0 \ suchthat \ f''(\eta_X) = 0 \ and \ f'(\eta) > 0 \ on \ [0, \eta_X]\}$$

$$Y = \{\delta < 0 / \text{there is first point } \eta_Y > 0 \ suchthat \ f''(\eta) < 0 \ and \ f'(\eta_Y) = 0 \ on \ [0, \eta_Y]\}$$

Two lemmas are presented below:

**Lemma 1.** *The set X is nonempty and open.*

**Proof.** Reconsideration of Equation (12) when $\eta = 0$, to get

$$(1 + C_1)\, f'''(0) = 1 - R(\lambda \delta - 1) - C_1 N'(0)$$

If $\delta = 0$, $f'''(0) = \frac{1}{1+C_1}(1 + R - C_1 N'(0))$.

As $N'(0) < 0$, we have $f'''(0) > 0$.

Therefore, we initially have $f' > 1$ and $f'' > 0$ on $(0, \varepsilon]$ for some $\varepsilon > 0$.

By continuity of the solutions of the IVP in its initial conditions for $\delta < 0$, sufficiently close to 0. $f'(\eta; \delta)$ will stay close to $f'(\eta; 0)$, i.e., will satisfy $f'(\eta, \delta) > 0$ on $[0, \varepsilon]$ with $f'(\varepsilon, \delta) > 1$. However, $f'(\eta, \delta)$ is less than 1 and decreasing for $\eta \in (0, \delta_1)$ for some $0 < \delta_1 < \varepsilon$. So for $f'$ to rise above 1, it must have a minimum. Thus, there exists a first point $\eta_x$, such that $f''(\eta_x, \delta) = 0$ with $f'(\eta, \delta) > 0$ on $[0, \eta_x]$. Thus for $\delta < 0$ sufficiently close to 0, we have $\delta \in X$.

Next for $X$ to be open, let $\overline{\delta} \in X$. We will show that all $\delta$ sufficiently close to $\overline{\delta}$ are in X.

At $\eta_x(\overline{\delta})$, we have $0 < f' < 1$ and $f'' = 0$.

The Equation (12) at $\eta_x(\overline{\delta})$ implies that

$$f'''(\eta_x) = \frac{1}{(1+C_1)}[R(f'(\eta_x))^2 + f'(\eta_x) - C_1 N'(\eta_x)] > 0$$

Thus by continuity of the solution of the IVP in the initial condition, for $\delta$ sufficiently close to $\overline{\delta}$, $f''(\eta, \delta) = 0$ will also have a root near $\eta_x(\delta)$ near $\eta_x(\overline{\delta})$ with $f'(\eta, \delta) > 0$. Thus $\delta \in X$. This leaves only the possibility that $f' = 0$ and $f'' = 0$, simultaneously; however, setting this information to Equation (12) yields $f'''(\eta) = -\frac{C_1}{1+C_1}N'(\eta)$ and $f'(\eta) = -\frac{C_1}{1+C_1}\int N(\eta)$, $\forall \eta$.

It contradicts to Equation (16), where $f' = 1$. □

**Lemma 2.** *The set Y is nonempty and open.*

By integrating Equation (12) on $[0, \eta]$, one is given

$$(1 + C_1) f''(\eta) = \delta + 2R \int_0^\eta (f'(t))^2 dt + R(\lambda - f(\eta)f'(\eta)) + f(\eta) - 1 - C_1 N(0) \qquad (17)$$

Now it is to show that there are values of $\delta < 0$ when magnitude of $\delta$ is large such that $f' = 0$ in the interval $(0, 1]$, say strictly before $f'' = 0$. Now suppose that this assertion is false and consider the following cases.

Case (1.a): $0 < f' < 1$, $f'' < 0$, for $\eta \in (0, 1]$, $\lambda \geq 0$ :.

Integrating $0 < f' < 1$ leads to $\lambda < f < \eta + \lambda - 1$ on $(0, 1]$, using these bounds in Equation (17) to get

$$f'' < \frac{\delta}{2} + 2R + R\lambda, \quad \eta \in (0, 1]$$

For a choice of $\delta < -2 - 4R - 2R\lambda$, results in $f'' < -1$ on $(0, 1]$, it shows $f'(0) < 0$, and it is in contradiction to the assumption that $f' > 0$ on $(0, 1]$.

Case (1.b): $0 < f' < 1$, $f'' < 0$, for $\eta \in (0, 1]$, $\lambda < 0$ :.

Integrating $0 < f' < 1$ leads to $\lambda < f < \eta + \lambda - 1$ on $(0, 1]$, using these bounds in Equation (17) to get

$$f'' < \frac{\delta}{2} + 2R, \quad \eta \in (0, 1]$$

For a choice of $\delta < -2 - 4R$, results in $f'' < -1$ on $(0, 1]$, it shows $f'(0) < 0$, and it is contradiction to the assumption that $f'(0) > 0$ on $(0, 1]$.

Case (2): suppose there exists $\eta_1 \in (0, 1]$ such that $f''(\eta_1) = 0$ with $f'' < 0$ on $(1, \eta_1)$.

Using bounds on $f''$ from Case (1), we have $f'' < \begin{cases} \frac{\delta}{2} + 2R + R\lambda & \text{if } \lambda \geq 0 \\ \frac{\delta}{2} + 2R & \text{if } \lambda < 0 \end{cases}$ for $\eta \in (0, \eta_1)$.

By choosing $\delta < \begin{cases} -(4R + 2R\lambda) & \text{if } \lambda \geq 0 \\ -4R & \text{if } \lambda < 0 \end{cases}$ , results in $f''(\eta_1) < 0$ contradicting $f''(\eta_1) = 0$.

Case (3): This leads to the possibility that $f'' = 0$ and $f' = 0$ simultaneously but as in Lemma 1, implying that $f'(\eta) = -\frac{C_1}{1+C_1}\int N(\eta)$, $\forall \eta$.

It contradicts to Equation (16), where $f' = 1$.

Thus $Y$ is non-empty. To show that $Y$ is open, let $\bar{\delta} \in Y$. Then there exists a first point $\eta_y(\bar{\delta})$ such that $f'(\eta_y(\bar{\delta})) = 0$ and $f''(\eta_y(\bar{\delta})) < 0$. By the continuity of the solution of the IVP in its initial conditions, for $\delta$ sufficiently close to $\bar{\delta}$, there exists $\eta_y(\delta)$ with $f'(\eta_y(\bar{\delta})) = 0$ and $f''(\eta_y(\bar{\delta})) < 0$ and, so, $Y$ is open.

Since the sets $X$ and $Y$ are disjoint, open and non-empty, there exists $\delta^*$, such that $\delta^* \notin X$ $\delta^* \notin Y$ and $X \cup Y \neq (-\infty, 0)$ where the interval $(-\infty, 0)$ is connected. Suppose $f' = 0$ and $f'' = 0$ simultaneously; thus, it can be concluded that $f'(\eta, \delta^*) > 0$ and $f''(\eta, \delta^*) < 0$ for all $\eta > 0$. Since $f'$ is bounded below and decreasing $f'(\infty, \delta^*) = C$ exists, where $0 \leq C < 1$.

We now show that we must have $C = 0$; for this, we suppose that $0 < C < 1$. Since $f''(\eta) < 0$ for $\eta > 0$, $f'$ is bound below by $C > 0$ and so $f$ tends to positive infinity. The term $ff''$ becomes negative. From Equation (12), we have

$$(1 + C_1)\, f'''(\eta) = f' + R((f'(\eta))^2 - ff'') - C_1 N'(\eta)$$

$$(1 + C_1)\, f'''(\eta) = f' + R(C^2 - ff'') - C_1 N'(\eta) \geq RC^2 = K_1 > 0, \text{ for } \eta \text{ large enough.}$$

There exists a point $\eta_2 > 0$ such that $\eta > \eta_2$, we have

$$(1 + C_1)\, f'''(\eta) > \frac{K_1}{2}$$

$$f'''(\eta) > \frac{K_1}{2(1 + C_1)}$$

On integration, we have $f''(\eta) > f''(\eta_2) + \frac{K_1}{2(1+C_1)}(\eta - \eta_2)$.

When $\eta \to \infty$ implies that $f'' \to \infty$, contradicting the fact that $f''(\eta) < 0$, so $f'(\infty, \delta^*) = 0$, it establishes the theorem.

Theorem: When $R > 0$ and $-\infty < \lambda < \infty$, there exists a solution to the boundary value problem to satisfy $f''(\eta) < 0$ and $f'(\eta) > 0$ for all $\eta > 0$.

## 4. Numerical Method

The finally resulted set of non-linear coupled ordinary differential equations is solved numerically subject to physical boundary conditions. Firstly, the third order derivative is reduced to second order by setting:

$$f' = q$$

The Equation (12) is transformed as below:

$$(1 + C_1)\, q'' + R(fq' - q^2) - q + C_1 N' = 0, \tag{18}$$

Suppose $\varphi$ represents each of the dependent functions: $f$, $q$, $N$, $\theta$ and then using central differences defined below:

$$\varphi'_n = \frac{\varphi_{n+1} - \varphi_{n-1}}{2h}$$

$$\varphi'_n = \frac{\varphi_{n+1} - \varphi_{n-1}}{2h}$$

$$\varphi''_n = \frac{\varphi_{n+1} - 2\varphi_n + \varphi_{n-1}}{h^2}$$

The second order differential Equations (13), (14) and (18) are now respectively approximated by central difference at typical grid point $\eta = \eta_n$ of the interval $[0, \infty)$. The discretization of the domain $[0, \infty)$ is uniform with a step size $h$. We obtain

$$
\begin{aligned}
\left[2(1 + C_1) + h^2(1 + q_n)\right] q_n &= \left[(1 + C_1) + \tfrac{Rh}{2} f_n\right] q_{n+1} \\
&+ \left[(1 + C_1) - \tfrac{Rh}{2} f_n\right] q_{n-1} + C_1 h (N_{n+1} - N_{n-1})
\end{aligned}
\tag{19}
$$

$$
\begin{aligned}
\left[1 + \tfrac{h}{2} C_3 f_n\right] N_{n+1} &- \left[2 + h^2 C_3 q_n - 2 C_1 C_2 h^2\right] N_n \\
&+ \left[1 - \tfrac{h}{2} C_3 f_n\right] N_{n-1} + C_1 C_2 \tfrac{h}{2} [q_{n+1} - q_{n-1}] = 0
\end{aligned}
\tag{20}
$$

$$
[1 + (1 - s) RP_r f_n h] \, \theta_{n+1} - 2\theta_n + [1 - (1 - s) RP_r f_n h] \, \theta_{n-1} = 0
\tag{21}
$$

The boundary conditions (15) become:

$$
\left.
\begin{aligned}
f_n(0) &= \lambda, \quad q_n(0) = 1, \quad N_n(0) = 0, \quad q_n(\infty) = 0 \text{ and } N_n(\infty) = 0, \\
\theta_n(0) &= 1, \, \theta_n(\infty) = 0, \qquad \text{Isothermal} \\
\theta'_n(0) &= -1, \, \theta'_n(\infty) = 0, \quad \text{Isoflux}
\end{aligned}
\right\}
\tag{22}
$$

The algebraic Equations (18)–(20) are solved iteratively by employing the SOR method (see Smith [39] p. 262) and the first order ODE (16) is integrated by Simpson's (1/3) rule Gerald [40] combined with a corrector formula (see Miline [41] p. 48) subject to the associated boundary conditions (21). The results for $f'(\eta)$, $N(\eta)$, $\theta(\eta)$, $-f''(0)$, $-N'(0)$ and $-\theta'(0)$ are in order of accuracy $o(h^2)$ because of the inherent second order approximation of central finite differences. The accuracy of the scheme is checked with repetition of the procedure at the grid size $h$, $h/2$ and $h/4$. The results obtained at three grid sizes are utilized to involve Richardson Extrapolation to the limit (see Burden [42] page 168) to achieve accuracy in the results in the order $o(h^6)$. The solution procedure, which is mainly based on the algorithm described in Syed et al. [43] is used to accelerate the iterative procedure and to improve the convergence. The relaxation parameter $\omega$ is optimized as proposed by Nakamura [44]. An initial guess is taken for $\omega$ as arbitrarily after each relaxation sweep except the first one, then the optimal values of $\omega$ are estimated with the formula given below:

$$
\omega^i_{opt} = \frac{2}{1 + \sqrt{1 - \left(\mu^i_J\right)^2}}
\tag{23}
$$

$$
\mu^i_\omega = \sqrt{\frac{N^i}{N^{i-1}}}, \quad \mu^i_J = \frac{\mu^i_\omega - 1 + \omega}{\omega \sqrt{\mu^i_\omega}}
\tag{24}
$$

and

$$
N^i = \sum_t \left(U^i_t - U^{i-1}_t\right)^2
\tag{25}
$$

where $i$ and $t$, respectively, stand for the grid point and relaxation sweep.

The process of optimization is stopped when the following criterion is met:

$$
\left| \mu^i_\omega - \mu^{i-1}_\omega \right| < E_{opt}
\tag{26}
$$

subject to the conditions $\mu^i_\omega < 1$ and $i > 1$. The iterative process is continued with $\omega = \omega_{opt}$ and $10^{-5} \le E_{opt} \le 10^{-3}$. The iterative procedure is stopped if the following criteria is satisfied for four consecutive iterations

$$
\begin{aligned}
&\max\left( \|q^{(t+1)} - q^{(t)}\|_2, \, \|N^{(t+1)} - N^{(t)}\|_2, \, \|f^{(t+1)} - f^{(t)}\|_2, \, \|\theta^{(t+1)} - \theta^{(t)}\|_2 \right) \\
&< TOL_{iter}
\end{aligned}
$$

here $TOL_{iter}$ is the prescribed error tolerance. The numerical scheme is coded in robust scientific programming language Fortron-90.

## 5. Results and Discussion

The theoretical and numerical analysis for this work was focused to reveal the physical nature of the micropolar fluids flow through porous medium caused by a permeable stretching sheet with both isothermal wall temperature condition and isoflux boundary condition. An extensive computational effort has been carried out to capture the reliable results for non-dimensional horizontal flow speed $f'(\eta)$, microrotation $N(\eta)$ and temperature function $\theta(\eta)$, skin friction coefficient $-f''(0)$, heat transfer rate at surface $-\theta'(0)$ and couple stress $-N'(0)$. The physical behavior of these quantities of interest has been scrutinized through the suitable ranges of the pertinent parameters—namely Reynolds number $R$, micropolar material parameter $C_1$, suction/injection parameter $\lambda$, Prandtl number Pr, heat index parameter $s$. The non-dimensional micropolar material parameters $C_1$, $C_2$ and $C_3$ are chosen arbitrarily. The values of $C_1$, are considered as 0.5, 1.0, 1.5 and 4.0, while $C_2 = 0.1$ and $C_3 = 0.5$ are chosen arbitrarily. However, if $C_1 = 0$ and $N = 0$, this problem corresponds to Newtonian fluids flow and heat transfer as studied by Dayyan et al. [24]. Calculations have been carried out for several values of all the above-mentioned parameters. The present findings show good agreement with previous studies [24], and hence prove their validity. In order to elaborate on this work, some representative outcomes are presented in tabular as well as in plot forms. Tables 1 and 2, respectively, present results for temperature function $\theta(\eta)$ with isothermal wall temperature and isoflux boundary conditions. The temperature function $\theta(\eta)$, decreases for micropolar fluid in comparison with that of Newtonian fluids. It is because that increase in the vortex viscosity of micropolar fluids provides a resistance to the thermal distribution. Table 3, demonstrates the impact of Reynolds number $R$ on $-f''(0)$, $-\theta'(0)$ and $-N'(0)$. Table 4 represents the iterative computations of temperature for three the grid sizes and its extrapolated values. Table 5 is presented to exhibit the efficiency of the iterative procedure, it enumerates the number of iterations and $\omega_{opt}$ for the grid sizes $h$, $h/2$, $h/4$ when the parameter $\lambda$ is varied. The increase in $R$ ($1 \leq R \leq 5$) marked an increase in the magnitude of these three physical quantities. These results show correspondence with previous studies by Dayyan et al. [24]. Moreover, the skin friction coefficient $-f''(0)$ is lesser in magnitude for micropolar fluids than that of Newtonian fluids; this outcome is supported by the experimental results of Fabula and Hoyt [45], but the heat transfer rate $-\theta'(0)$ at the surface is greater in magnitude for micropolar fluids than that of Newtonian fluids.

**Table 1.** The results of $\theta$ (for isothermal wall temperature) when $\lambda = 0$, $\mathrm{Pr} = 1$, $s = 0$ and $R = 1$.

| $\eta$ | Newtonian Fluids | | Micropolar Fluids | | |
| --- | --- | --- | --- | --- | --- |
| | Dayyan [24] | Present Results | $C_1 = 0.5$ | $C_1 = 1.5$ | $C_1 = 4.0$ |
| 0.0 | 1 | 1 | 1 | 1 | 1 |
| 0.2 | 0.89996 | 0.89996 | 0.89052 | 0.88023 | 0.86945 |
| 0.4 | 0.80327 | 0.80327 | 0.78480 | 0.76471 | 0.74366 |
| 0.6 | 0.71225 | 0.71225 | 0.68559 | 0.65665 | 0.62643 |
| 0.8 | 0.62826 | 0.62826 | 0.59456 | 0.55813 | 0.52030 |
| 1.0 | 0.55191 | 0.55192 | 0.51253 | 0.47021 | 0.42665 |
| 1.2 | 0.48329 | 0.48330 | 0.43964 | 0.39315 | 0.34584 |
| 1.4 | 0.42214 | 0.42215 | 0.37561 | 0.32660 | 0.27745 |
| 1.6 | 0.36800 | 0.36801 | 0.31986 | 0.26982 | 0.22054 |
| 1.8 | 0.32029 | 0.32030 | 0.27166 | 0.22188 | 0.17387 |
| 2.0 | 0.27841 | 0.27842 | 0.23023 | 0.18175 | 0.13609 |
| 2.2 | 0.24175 | 0.24175 | 0.19476 | 0.14838 | 0.10583 |
| 2.4 | 0.20971 | 0.20972 | 0.16452 | 0.12080 | 0.08182 |
| 2.6 | 0.18177 | 0.18178 | 0.13879 | 0.09812 | 0.06294 |
| 2.8 | 0.15743 | 0.15743 | 0.11696 | 0.07954 | 0.04819 |
| 3.0 | 0.13623 | 0.13624 | 0.09847 | 0.06436 | 0.03673 |

**Table 2.** The results of $\theta$ (for isoflux boundary) when $\lambda = 0$, $\mathrm{Pr} = 1$, $s = 1$ and $R = 1$.

| $\eta$ | Newtonian Fluids | | Micropolar Fluids | | |
|---|---|---|---|---|---|
| | Dayyan [24] | Present Results | $C_1 = 0.5$ | $C_1 = 1.5$ | $C_1 = 4.0$ |
| 0.0 | 1.9615 | 1.96687 | 1.81535 | 1.65930 | 1.52159 |
| 0.2 | 1.7627 | 1.76810 | 1.61660 | 1.46057 | 1.32287 |
| 0.4 | 1.5706 | 1.57599 | 1.42470 | 1.26888 | 1.13141 |
| 0.6 | 1.3898 | 1.38515 | 1.24459 | 1.08957 | 0.95297 |
| 0.8 | 1.2229 | 1.22828 | 1.07934 | 0.92610 | 0.79143 |
| 1.0 | 1.0712 | 1.07658 | 0.93042 | 0.78023 | 0.64891 |
| 1.2 | 0.9340 | 0.93024 | 0.79810 | 0.65236 | 0.52593 |
| 1.4 | 0.8134 | 0.81875 | 0.68186 | 0.54192 | 0.42187 |
| 1.6 | 0.7058 | 0.70118 | 0.58066 | 0.44771 | 0.33529 |
| 1.8 | 0.6110 | 0.61639 | 0.49316 | 0.36817 | 0.26431 |
| 2.0 | 0.5278 | 0.52318 | 0.41794 | 0.30157 | 0.20686 |
| 2.2 | 0.4549 | 0.45033 | 0.35357 | 0.24621 | 0.16087 |
| 2.4 | 0.3913 | 0.39669 | 0.29866 | 0.20045 | 0.12442 |
| 2.6 | 0.3358 | 0.33117 | 0.25196 | 0.16281 | 0.09575 |
| 2.8 | 0.2874 | 0.28280 | 0.21233 | 0.13197 | 0.07338 |
| 3.0 | 0.2453 | 0.24069 | 0.17876 | 0.10679 | 0.05602 |

**Table 3.** The results of skin friction $f''(0)$, heat transfer rate $\theta(0)$, couple stress $N'(0)$ when $\lambda = 0$, $C_1 = 0.5$, $\mathrm{Pr} = 1$ and $s = 0$ (isothermal).

| $R$ | Newtonian Fluids | Micropolar Fluids | Newtonian Fluids | Micropolar Fluids | Micropolar Fluids |
|---|---|---|---|---|---|
| | $-f''(0)$ | | $-\theta'(0)$ | | $-N'(0)$ |
| 1.0 | 1.41422 | 1.15132 | 0.50331 | 0.55086 | 0.05390 |
| 2.0 | 1.73205 | 1.41089 | 0.76007 | 0.82297 | 0.05995 |
| 5.0 | 2.44949 | 1.33690 | 1.35067 | 1.25761 | 0.0694 |

**Table 4.** The results of $\theta$ (for isothermal wall temperature) when $\lambda = 0$, $\mathrm{Pr} = 1$, $s = 0$, $C_1 = 0.5$ and $R = 1$.

| $\eta$ | $h$ | $h/2$ | $h/4$ | Extapolated |
|---|---|---|---|---|
| 0.0 | 1 | 1 | 1 | 1 |
| 0.2 | 0.87541 | 0.88534 | 0.88745 | 0.89052 |
| 0.4 | 0.77238 | 0.77503 | 0.78087 | 0.78480 |
| 0.6 | 0.67225 | 0.67925 | 0.68102 | 0.68559 |
| 0.8 | 0.58662 | 0.58921 | 0.59073 | 0.59456 |
| 1.0 | 0.50147 | 0.50591 | 0.50918 | 0.51253 |
| 1.2 | 0.42220 | 0.42916 | 0.43412 | 0.43964 |
| 1.4 | 0.36514 | 0.36926 | 0.37217 | 0.37561 |
| 1.6 | 0.31109 | 0.31329 | 0.31701 | 0.31986 |
| 1.8 | 0.27029 | 0.27060 | 0.27104 | 0.27166 |
| 2.0 | 0.22835 | 0.22987 | 0.23002 | 0.23023 |
| 2.2 | 0.18771 | 0.18936 | 0.18132 | 0.19476 |
| 2.4 | 0.14971 | 0.15572 | 0.15934 | 0.16452 |
| 2.6 | 0.13177 | 0.13219 | 0.13453 | 0.13879 |
| 2.8 | 0.10743 | 0.11066 | 0.11339 | 0.11696 |
| 3.0 | 0.09223 | 0.09362 | 0.09621 | 0.09847 |

**Table 5.** Number of Iterations $N_1$ and Optimum Value of Relaxation Parameter $\omega_{opt}$ in SOR Method.

| $\lambda$ | $h_1 = 0.1$ | | $h_2 = 0.05$ | | $h_3 = 0.025$ | |
|---|---|---|---|---|---|---|
| | $N_1$ | $\omega_{opt}$ | $N_1$ | $\omega_{opt}$ | $N_1$ | $\omega_{opt}$ |
| −0.3 | 45 | 1.60 | 94 | 1.65 | 243 | 1.70 |
| −0.1 | 42 | 1.60 | 108 | 1.65 | 211 | 1.70 |
| 0.0 | 37 | 1.60 | 81 | 1.65 | 189 | 1.70 |
| 0.1 | 52 | 1.60 | 122 | 1.65 | 231 | 1.70 |
| 0.3 | 57 | 1.60 | 137 | 1.65 | 263 | 1.70 |

Figure 2 illustrates the reduction in flow speed $f'(\eta)$ with increments in $R$. Figure 3 shows that the flow speed $f'(\eta)$ is increased when injection parameter $\lambda$ ($\lambda < 0$) increases, this result corresponds to physical nature of the injection/suction parameter. The increasing values of the micropolar parameter $C_1$, marked as significant in $f'(\eta)$ and increment in boundary layer thickness as well as shown Figure 4. This result attracts our special attention, as it reveals the clear difference between the dynamics of Newtonian fluids and micropolar fluids. Here, the curve for $C_1 = 0$ corresponds to Newtonian fluid flow.

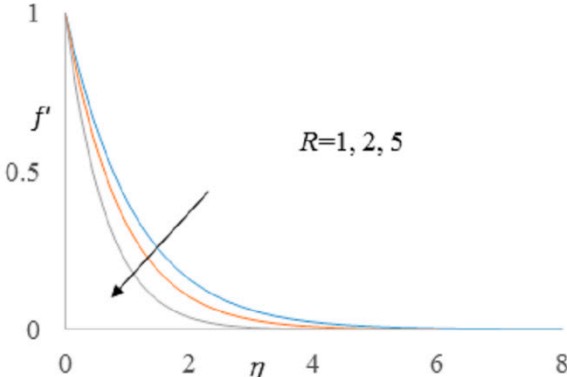

**Figure 2.** Graph of $f'$ (isothermal) with variation of $R$.

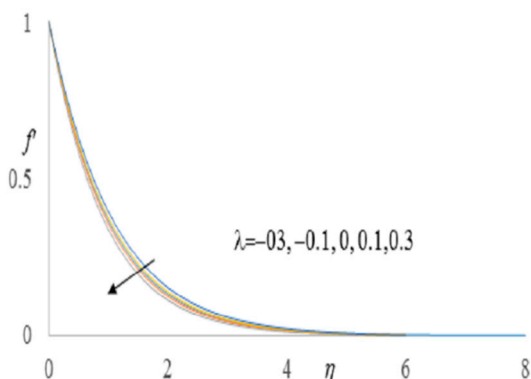

**Figure 3.** Graph of $f'$ (isothermal) with variation of $\lambda$.

Figures 5 and 6 are presented, respectively, to display the non-dimensional microrotation $N(\eta)$ under the effects of parameters $R$ and $\lambda$. It is revealed that $N(\eta)$ increases near the boundary and then swings down away from the boundary when $R$ and $\lambda$ ($\lambda > 0$) are increased. Figure 7 shows that the increment in the values of micropolar material parameter $C_1$ shows a notable rise in the curve of $N(\eta)$. It is reasonable from the physical point of view, that the increment in vortex viscosity causes an increase in the microrotation of the microstructures suspended in the flow field.

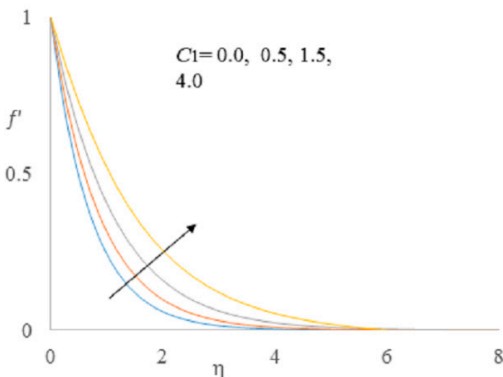

**Figure 4.** Graph of $f'$ (isothermal) with variation of $C_1$.

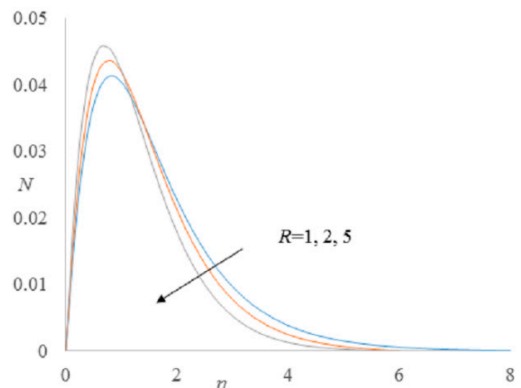

**Figure 5.** Graph of $N$ (isothermal) with variation of $R$.

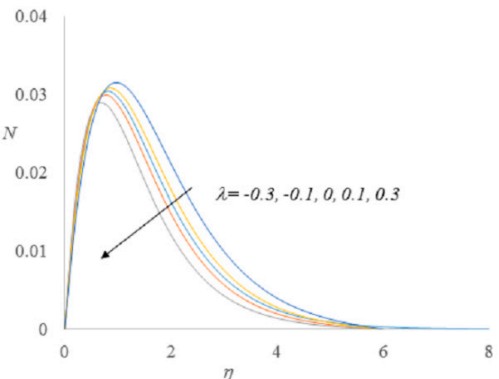

**Figure 6.** Graph of $N$ (isothermal) with variation of $\lambda$.

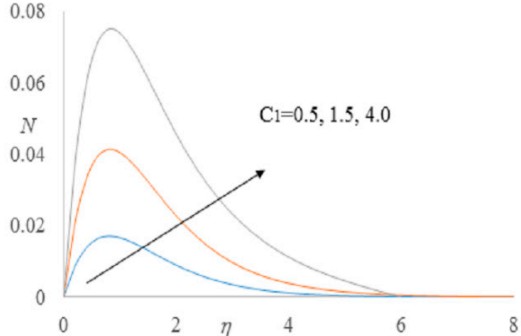

**Figure 7.** Graph of $N$ (isothermal) with variation of $C_1$.

Figures 8–11 demonstrate the decreasing pattern of temperature function $\theta(\eta)$, with increasing values of the parameters Pr, $\lambda$, $C_1$ and $n$, respectively (for isothermal wall temperature boundary conditions). The increase in the value of Pr, means a decrease in thermal conductivity and a reduction in thermal distribution as revealed in Figure 8. Furthermore, Figure 9 shows that the injection $\lambda$ ($\lambda < 0$) raises the temperature function $\theta(\eta)$, but suction $\lambda$ ($\lambda > 0$) causes a reduction in $\theta(\eta)$. The increase in heat index parameter $s$ ($0 \leq s \leq 10$) made a marked reduction in temperature distribution and, hence, a decrease in the thermal boundary layer is observed as depicted in Figure 10. Furthermore, Figure 11 shows that the temperature function $\theta(\eta)$ becomes less for micropolar fluids than that of Newtonian fluids. This is because of the pressure of spinning micro structures in micropolar fluids provide resistance to heat distribution in the bulk fluid.

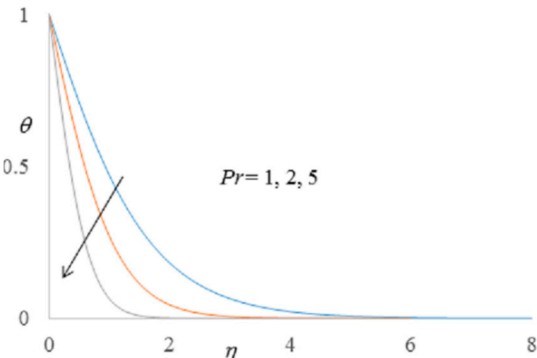

**Figure 8.** Graph of $\theta$ (isothermal) with variation of Pr.

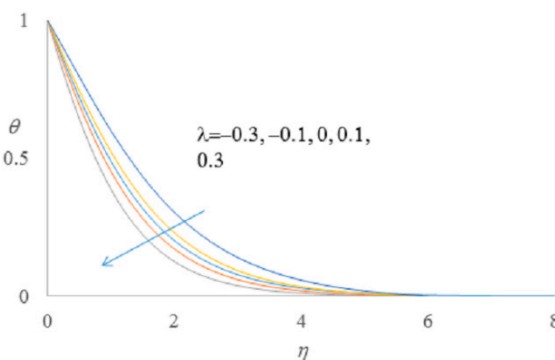

**Figure 9.** Graph of $\theta$ (isothermal) with variation of $\lambda$.

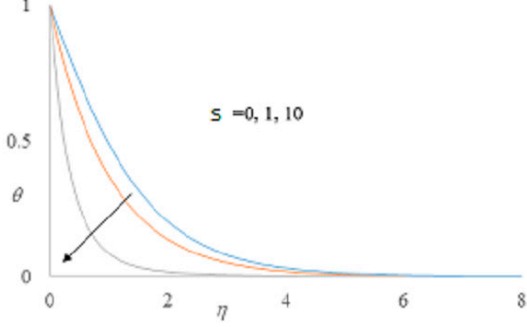

**Figure 10.** Graph of $\theta$ (isothermal) with variation of s.

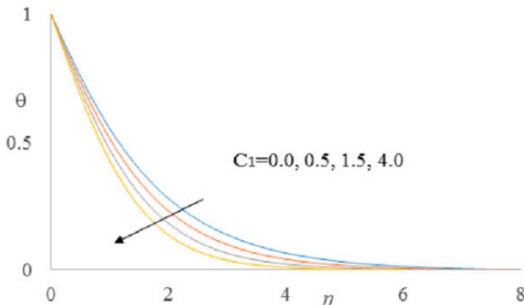

**Figure 11.** Graph of $\theta$ (isothermal) with variation of $C_1$.

The results of $\theta(\eta)$, for isoflux boundary conditions have been computed and presented, respectively, in Figures 12–15 under the influence of Pr, $R$, $\lambda$ and $C_1$. The increment in Pr, $R$, indicates a reduction in $\theta(\eta)$, respectively, in Figures 12 and 13. This is because of Pr , which is reciprocal to thermal conductivity, and $R$ is reciprocal to the thermal porosity of the medium. Figure 14 illustrates that an increment in injection $\lambda$ ($\lambda < 0$) causes a decrease in $\theta(\eta)$, but an increment in suction at the wall $\lambda$ ($\lambda > 0$) raises the curve of $\theta(\eta)$. The heat function $\theta(\eta)$ is greater for Newtonian fluids than that of micropolar fluids as depicted in Figure 15.

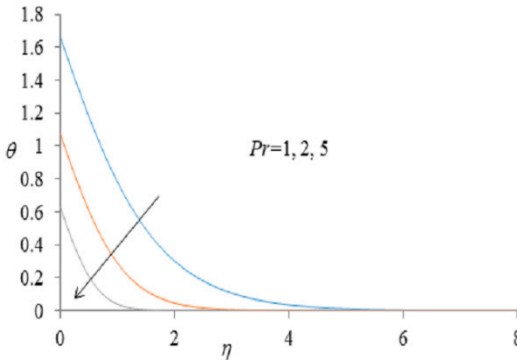

**Figure 12.** Graph of $\theta$ (isoflux) with variation of Pr.

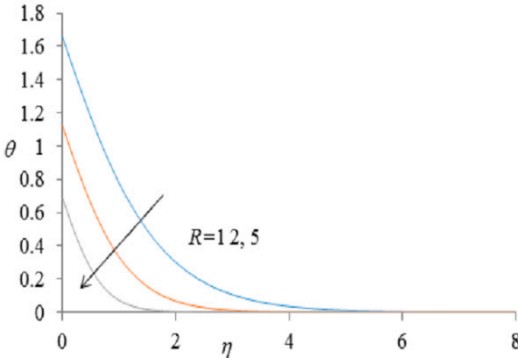

**Figure 13.** Graph of $\theta$ (isoflux) with variation of $R$.

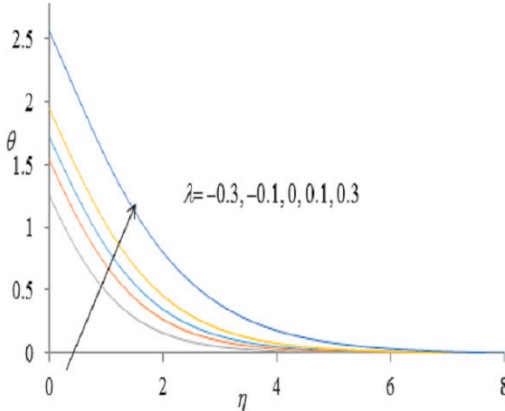

**Figure 14.** Graph of $\theta$ (isoflux) with variation of $\lambda$.

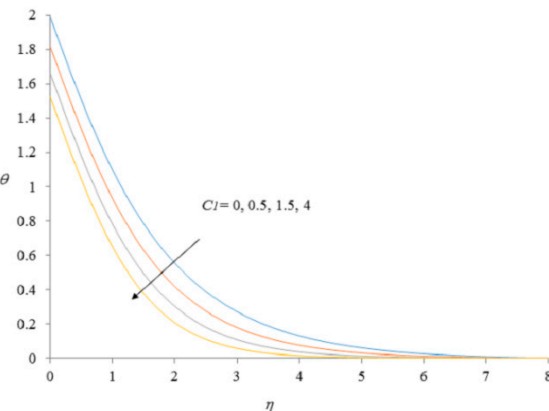

**Figure 15.** Graph of $\theta$ (isoflux) with variation of $C_1$.

## 6. Conclusions

The free convection boundary layer flow of micropolar fluids through porous medium over a permeable stretching sheet with two thermal boundary conditions which are either power law heat flux or wall temperature is analyzed numerically. The theoretical inclusion of spinning of a suspended microstructure in the bulk flow requires the coupling of angular momentum with linear momentum. Similarly, both the first and second laws of thermodynamics have been applied in heat transfer analysis. A reliable, efficient numerical scheme based on finite difference approximation has been applied with improved accuracy to be achieved through the extrapolation technique. The novelty of the present results included a reduction in flow speed, temperature distribution, skin friction coefficient, and an increase in the heat transfer rate. Moreover, the output in this work presents the results for microrotation and couple stress. The main outcomes of this research work are as follows:

The velocity in the boundary layer is retorted against enhanced values of the Reynold's number $R$, as well as that of suction $\lambda < 0$.

The velocity is incremented and the trend of flow becomes faster, increasing the strength of micropolar parameter $C_1$ or injection ($\lambda > 0$). The microrotation is incremented with higher values of $C_1$. The microrotation exhibits an oscillatory pattern as it increases near the surface and then decreases away from the surface with increasing values of $R$ and $\lambda$.

The temperature function $\theta(\eta)$ and thermal boundary layer thickness decrease against Pr, $R$, $\lambda$, $s$ and $C_1$ for both cases; isothermal, isoflux, and wall temperature conditions.

Skin friction coefficient $-f''(0)$ and local Nusselt number $-\theta'(0)$ are smaller in value for micropolar fluids than for Newtonian fluids.

Couple stress $-N'(0)$ increases with an increase in $C_1$.

In short, the present findings embrace microrotation and micropolar parameters to reveal the real nature of fluid motion and thermal distribution. The varying values of micropolar parameter $C_1$ indicate a notable impact on the physical nature of flow and temperature transportation. In this sense, the present work is more practicable than the previous studies for Newtonian fluids. The numerical scheme used herein is simple and efficient but it may not work for heat and mass transfer across any complex geometry. The numerical solution for the basic partial differential equation may provide more versatile results than the transformed model.

**Author Contributions:** Conceptualization, S.H.; Investigation, F.A.; Methodology, F.A.; Software, A.O.A.; Validation, S.E.F.; Writing—original draft, A.O.A. and S.E.F.; Writing—review & editing, R.U. All authors have read and agreed to the published version of the manuscript.

**Funding:** This research received no external funding.

**Acknowledgments:** The authors are thankful to NTU, Singapore, Mathematics Department, GC University Lahore and UOH, KSA for providing research environment.

**Conflicts of Interest:** The authors declare no conflict of interest.

**Availability of Data and Material:** All data is available and no need of any permission etc.

## Nomenclature

| | |
|---|---|
| $\lambda$ | Injection parameter |
| $K$ | Permeability of the porous medium |
| $R$ | Reynolds number |
| $u$ | Velocity in direction |
| $U_0$ | Wall velocity coefficient |
| $v$ | Velocity in direction |
| $v_w$ | Injection velocity |
| $x$ | Coordinate system |
| $y$ | Coordinate system |
| $\rho$ | Density of the fluid |
| $\Psi$ | Stream function |
| $\mu$ | Dynamic viscosity |
| $\kappa$ | Vortex Viscosity |
| $j$ | Micro Inertia |
| $\gamma$ | Spin Viscosity |
| $\lambda_1$ | Stokes Viscosity |
| $\eta$ | Dimensionless similarity variable |
| $\nu$ | Kinematic viscosity |
| $C_1$, $C_2$, $C_3$ | Dimensionless Material Constants |
| $N$ | Dimensionless similarity function for microrotation |
| $T$ | Fluid Temperature |
| $\theta$ | Nondimensional Temperature |
| $\Omega$ | Microrotation |
| $N$ | Nondimensional Microrotation |
| $s$ | Heat Index Parameter |
| $L$ | Length of porous plate |
| $\alpha_{eff}$ | is the effective thermal diffusivity |
| $\alpha$, $\beta$, $\lambda_1$ | additional viscosity coefficients |
| Pr | Prandtl number |
| $\partial^2 u/\partial x^2 + \partial^2 u/\partial y^2$ | Diffusion term |
| $u\partial u/\partial x + v\partial u/\partial y$ | Convection term |

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
