# Peer review of "Numerical Solution of Nonlinear Diff. Equations for Heat Transfer in Micropolar Fluids over a Stretching Domain"

_mathematics, doi:10.3390/math8050854_

Round 1
Reviewer 1 Report
This work presented a numerical solution for the flow and heat transfer of micropolar fluid through a porous medium. The topic is suitable for the journal. Some interesting results were presented. It can be accepted for publication after addressing the following issues.
- The cited references should be organized in a sequence of their presences. Please revise.
- The consistent format should be used throughout the manuscript. For example, the font of section title and main contents. Tables and figures should be placed close to the discussion. Please avoid stacking them together for better reading and visualization.
- The title is lengthy. Please revise it concisely.
- The literature review is inadequate without an explicit discussion of the gap between existing work and the current work. In other words, what are the motivations of the current work? What is the usefulness of the presented solution in real applications?
- The scientific value of the current work should be further discussed. What is the fundamental understanding of observations from the calculation? How did authors validate their findings, from experiment or comparison to existing works?
- The solution was implemented with iterative calculations. Therefore, it will be interesting to discuss the computational efficiency of the presented solutions.
Author Response
Authors’ Response to Comments of Reviewer 1
Authors are deeply thankful to the learned reviewer for sparing time to review the manuscript and making valuable comments. The authors’ response to each of the comments is given below:
- Comment #1: The cited references should be organized in a sequence of their presences. Please revise.
Authors’ response: Agreed, the references have been organized in sequence of their presence.
- Comment #2: The consistent format should be used throughout the manuscript. For example, the font of section title and main contents. Tables and figures should be placed close to the discussion. Please avoid stacking them together for better reading and visualization.
Authors’ response: Agreed, the advised suggestions are considered. A consistent formatting of the manuscript is made to present better reading and visualization.
- Comment #3: The title is lengthy. Please revise it concisely.
Authors’ response: According to the wise suggestion of the expert reviewer, the Title is now made shorter. Revised Title is “Numerical Solution of Coupled Diff. Equations for Heat Transfer in Micropolar Fluids over a Stretching Domain”.
- Comment #4: The literature review is inadequate without an explicit discussion of the gap between existing work and the current work. In other words, what are the motivations of the current work? What is the usefulness of the presented solution in real applications?
Authors’ response: This is really an important point to be addressed. The motivation of current work along with usefulness of presented solution is described now in last paragraph of Introduction Section.
- Comment #5: The scientific value of the current work should be further discussed. What is the fundamental understanding of observations from the calculation? How did authors validate their findings, from experiment or comparison to existing works?
Authors’ response: This aspect of the study is rightly pointed out by the worthy reviewer. Surely, there are several numerical methods and various built in numerical schemes in commercially available computational softwares. Most of these computational facilities lack in the justification of their accuracy. We employed finite difference based numerical scheme with utilization of Successive Over Relaxation (SOR) Method, Simpson’s (1/3) rule. Accuracy for this solution is assured through their computations for three grids sizes (h, h/2, h/4) and their closeness with the results of related published studies in the limiting case. In addition, extrapolation routine raises the order of accuracy to higher level. Table 4 indicates, how the results are rectified with reduction of step size and extrapolation.
Moreover, this work is attempted to develop better understanding of the micropolar structure of viscous fluids for the flow and thermal characteristics in porous medium over a permeable stretching sheet, we observed, heat transfer rate, shear and couple stresses at the sheet surface and the velocity, microrotation and temperature fields in the boundary layers over the sheet. It is noticed that Skin friction coefficient -f′′(0) and local Nusselt number -θ′(0) are lesser in value for micropolar fluids than for Newtonian fluids. Couple stress -N′(0) increases with increase in C₁( micropolar parameter)
Comment #6: The solution was implemented with iterative calculations. Therefore, it will be interesting to discuss the computational efficiency of the presented solutions.
Authors’ response: The computational code is developed and run in the environment of ForTran-90 on a simple desktop computer of Pantium 3 model. The iterative procedure worked very efficiently. The solution converged after small number of iterations. In compliance of the wise suggestion, Table 5 is now presented to show the enumeration of iteration with variation of step size and suction injection parameter. The selection of optimum value of relaxation parameter also played its role to increase efficiency of the iterative procedure.
On behalf of all authors
First & Corresponding Author
Prof. Dr. Farooq Ahmad
School of Mechanical and Aerospace Engineering,
Nanyang Technological University, Singapore
ahmad.farooq@ntu.edu.sg

Reviewer 2 Report
In these paper Authors presented a numerical study based on finite difference approximation to analyse the bulk flow, micro spin flow and heat transfer phenomenon for micropolar fluids dynamics through Darcy porous medium. Authors considered fluid flow mechanism over a moving permeable sheet. On the basis of porosity of medium, similarity functions are utilized to avail a set of ordinary differential equations. Below I presented some remarks that came to my mind during reading:
- The paper must be necessarily adapted to Mathematics template.
- References must be numbered in the order they appear in the text.
- In my opinion the Introduction must be improved. Introduction should clearly specify the purpose and motivation of taking up the topic. The Authors should state what has justified using the given method, what is special, unexpected, or different in their approach. The gap between previous studies and present studies should be strongly emphasized. Please also organize the introduction section as this order: importance and meaning, previous studies (literature review), the gap between previous studies and present studies, objectives. I consider that the manuscript under review will benefit if the authors make all of these aspects as clear as possible to the readers.
- Chapter 4. Results and Discussion: Could the authors compare their method from the manuscript and other ones that have been developed and used in the literature for the same purpose in more detail? Moreover, in this section Authors should also highlight what are the advantages and disadvantages when comparing their devised solution with other solutions from the scientific literature.
- In Conclusions Authors must try to emphasize novelty, put some quantifications, and comment on the limitations. This is a very common way to write conclusions for academic journal. The conclusions should highlight the novelty and advance in understanding presented in the work.
- "Authors Contributions" and "Conflict of Interest" are missing.
- References should be prepared in accordance with the Mathematics template.
Author Response
Authors’ Response to Comments of Reviewer 2
Authors are deeply thankful to the learned reviewer for sparing time to review the manuscript and making valuable comments. The authors’ response to each of the comments is given below:
Comment #1: The paper must be necessarily adapted to Mathematics template.
Authors’ response: Agreed, The manuscript is now reformatted to adapt Mathematics template
Comment #2: References must be numbered in the order they appear in the text.
Authors’ response: Agreed, the references have been organized in sequence of their presence in the text.
Comment #3: In my opinion the Introduction must be improved. Introduction should clearly specify the purpose and motivation of taking up the topic. The Authors should state what has justified using the given method, what is special, unexpected, or different in their approach. The gap between previous studies and present studies should be strongly emphasized. Please also organize the introduction section as this order: importance and meaning, previous studies (literature review), the gap between previous studies and present studies, objectives. I consider that the manuscript under review will benefit if the authors make all of these aspects as clear as possible to the readers.
Authors’ response: This is surely an important point to be addressed. The motivation of current work along with fulfillment of other suggestions is described now in last paragraph of Introduction Section.
Comment #4: Chapter 4. Results and Discussion: Could the authors compare their method from the manuscript and other ones that have been developed and used in the literature for the same purpose in more detail? Moreover, in this section Authors should also highlight what are the advantages and disadvantages when comparing their devised solution with other solutions from the scientific literature.
Authors’ response: This aspect of the study is rightly pointed out by the worthy reviewer. Surely, there are several numerical methods and various built in numerical schemes in commercially available computational softwares. Most of these computational facilities lack in the justification of their accuracy. We employed finite difference based numerical scheme with utilization of Successive Over Relaxation (SOR) Method, Simpson’s (1/3) rule. Accuracy for this solution is assured through their computations for three grids sizes (h, h/2, h/4) and their closeness with the results of related published studies in the limiting case. In addition, extrapolation routine raises the order of accuracy to higher level. Table 4 indicates, how the results are rectified with reduction of step size and extrapolation.
Comment #5: In Conclusions Authors must try to emphasize novelty, put some quantifications, and comment on the limitations. This is a very common way to write conclusions for academic journal. The conclusions should highlight the novelty and advance in understanding presented in the work.
Authors’ response: In compliance with this justified comment, we revised the conclusion section and tried to make it sound as per suggestions of the reviewer. In addition to over all revision of conclusion, the following lines are included to show the novelty and future work.
“In short, the present findings embrace microrotation and micropolar parameters to reveal the real nature of the fluid motion and thermal distribution. The varying values of micropolar parameter C₁ indicated notable impact on physical nature of flow and temperature transportation. In this sense, the present work is more practicable than the previous studies for Newtonian fluids. The numerical scheme used herein is simple and efficient but it may not work for heat and mass transfer across any complex geometry. Numerical solution for basic partial differential equation may provide versatile results than the transformed model”.
Comment #6: "Authors Contributions" and "Conflict of Interest" are missing.
Authors’ response: "Authors Contributions" and "Conflict of Interest" are now added.
Comment #7: References should be prepared in accordance with the Mathematics template.
Authors’ response: Agreed, the references have been prepared in accordance with the Mathematics template accordingly.
On behalf of all authors
First & Corresponding Author
Prof. Dr. Farooq Ahmad
School of Mechanical and Aerospace Engineering,
Nanyang Technological University, Singapore
ahmad.farooq@ntu.edu.sg

Reviewer 3 Report
In the paper, the authors study the bulk flow, micro spin flow and heat transfer for micropolar fluids dynamics through Darcy porous medium. The corresponding non-linear equations are solved by a numerical scheme that involves Simpson’s rule and the method of successive over-relaxation. The authors cite extensive literature, substantiate the importance of the considered model in technological processes, quite clearly formulate the problem under consideration and discuss in detail the results obtained. These results will undoubtedly be of interest to specialists in the field of hydrodynamics and its applications. However, the mathematical contribution of the paper is small. In fact, the authors apply well-known numerical methods to a system of coupled ordinary differential equations. I do not see serious mathematical difficulties that the authors would have to overcome, as well as new mathematical ideas proposed in the work. Although I emphasize that in general the paper is interesting.
In my opinion, the authors should submit this manuscript to a specialized journal related to mathematical modeling of physical processes or a journal in mechanics fields. A good option would be a multidisciplinary journal such as Symmetry.
Conclusion: I do not recommend this paper for publication in Mathematics.
Author Response
Authors’ Response to Comments of Reviewer 3
Authors are highly thankful to the learned reviewer for sparing time for thorough review of the manuscript and making valuable comments. We also feel encouraged with section wise appreciation by the honorable reviewer.
Comment #1: “these results will undoubtedly be of interest to specialists in the field of hydrodynamics and its applications. However, the mathematical contribution of the paper is small”,
Authors’ Response: Thanks for acknowledging that the results are of interests. The aspects of the study are rightly pointed out by the worthy reviewer. We are not challenging the opinion of the worthy reviewer but we are just explaining our a little participation in the paper. Surely, there are several numerical methods and various built in numerical schemes in commercially available computational softwares. Most of these computational facilities lack in the justification of their accuracy. We employed finite difference based numerical scheme with utilization of Successive Over Relaxation (SOR) Method, Simpson’s (1/3) rule. Accuracy for this solution is assured through their computations for three grids sizes (h, h/2, h/4) and their closeness with the results of related published studies in the limiting case. In addition, extrapolation routine raises the order of accuracy to higher level. Table 4 indicates, how the results are rectified with reduction of step size and extrapolation.
Moreover, this work is attempted to develop better understanding of the micropolar structure of viscous fluids for the flow and thermal characteristics in porous medium over a permeable stretching sheet, we observed, heat transfer rate, shear and couple stresses at the sheet surface and the velocity, microrotation and temperature fields in the boundary layers over the sheet. It is noticed that Skin friction coefficient -f′′(0) and local Nusselt number -θ′(0) are lesser in value for micropolar fluids than for Newtonian fluids. Couple stress -N′(0) increases with increase in C₁( micropolar parameter).
Comment #2: In my opinion, the authors should submit this manuscript to a specialized journal related to mathematical modeling of physical processes or a journal in mechanics fields. A good option would be a multidisciplinary journal such as Symmetry.
Conclusion: I do not recommend this paper for publication in Mathematics.
Authors’ Response: The honorable reviewer’s suggestion is right 100% right, we honor that opinion.
Kindly hear our explanation: actually the paper under review titled “Numerical Solution of Non-Linear Differential Equations Governing the Flow and Heat Transfer of Micropolar Fluids in a Porous Stretching Domain” was presented in World Congress ICIAM 2019 physically as oral talk by the undersigned (corresponding author of the paper) in July 2019, Valencia, Spain. Certificate of participation is attached/pasted below.
More over the journal “Mathematics” announced special issue tiled “Selected Papers from Iterative Processes for Solving Nonlinear Problems: Convergence and Stability of ICIAM 2019 and MME&HB 2019”, so the authors submitted the paper in the special issue meant for selected papers from ICIAM 2019.
Kindly allow us to carry on for the special issue tiled “Selected Papers from Iterative Processes for Solving Nonlinear Problems: Convergence and Stability of ICIAM 2019 and MME&HB 2019”, and not for any regular issue of “Mathematics”. We have amended title of the article now on the advises of the other reviewer i.e. “Numerical Solution of Coupled Diff. Equations for Heat Transfer in Micropolar Fluids over a Stretching Domain”
We are much obliged about the advises and comments of the reviewer.
On behalf of all authors
First & Corresponding Author
Prof. Dr. Farooq Ahmad
School of Mechanical and Aerospace Engineering,
Nanyang Technological University, Singapore
Ahmad.farooq@ntu.edu.sg

Round 2
Reviewer 1 Report
The authors have addressed some of the raised issues. However, the authors should provide a detailed response letter to clarify whether the authors agree with the comments and how the comments are addressed in the revised manuscript.
Author Response
Authors’ Responses
We are highly obliged to the honorable reviewer for critical review of the manuscript and delivering detailed suggestions. We are strongly agreed with all the raised issues by the expert reviewer and we have addressed these issues accordingly. The detail of responses to each comment is presented. It is to mention that the manuscript was drafted in “Scienific Work place”, it is a recommended plaform for scientific document preparation. Comment# 1 is correctly made, there were few unarranged references, it was actually the same difficulty as discussed above. SWP offers its own way of setting References, it created problem in few references. The references are organized in sequence of their presence and these are cross checked. Now there is no such issue and also the manuscript has been converted in word format as advised by the editorial representative so a different style than that of Scientific work place format.
- In response to the comment#2, it is submitted that a consistent formatting of the manuscript is made to present better reading and visualization. The overall outlook of the manuscript was poor. It is rightly observed by the wise reviewer. We have now rewritten the manuscript in Ms Word. There is consistency in document setting. As far as the comment#3 is concerned, it is rightly judged in the review processes. The revised concise Title is “Numerical Solution of Coupled Diff. Equations for Heat Transfer in Micropolar Fluids over a Stretching Domain”. Response to Comment #4 : The inadequacy of literature review and lack of motivation are bring to light. We agreed with these important points. The introduction section has been revised and updated accordingly. An integral part on inherited Non-linear nature of flow problems is introduced with reference of working solution procedures in the existing literature. We make our point clear that why we used finite difference based numerical scheme. It works well for simple geometries and simple grids. In this case its efficiency, accuracy and convergence are acceptable. Moreover, the sources distantly related to the problem as pointed out by the worthy reviewer are removed with agreement to the wise suggestion. Motivation of work is also discussed.
Comment #5: The scientific value of the current work should be further discussed. What is the fundamental understanding of observations from the calculation? How did authors validate their findings, from experiment or comparison to existing works?
Authors’ response: The comment is justified. This aspect of the study is rightly pointed out by the worthy reviewer. An integral part on inherited Non-linear nature of flow problems is introduced with reference of working solution procedures in the existing literature. We make our point clear that why we used finite difference based numerical scheme. A new section on “Existence and Uniqueness Results” is added. To best of our knowledge, it is first solvability attempt about this type of flow problem of micropolar fluids. It has added to the quality of manuscript. An effeort has been made to strengthen the mathematical analysis of the problem. The section on Mathematical Analysis is improved sufficiently. Moreover, the section on Numerical Method is also improved in Mathematical terms. Surely, there are several numerical methods and various built in numerical schemes in commercially available computational softwares. Most of these computational facilities lack in the justification of their accuracy. We employed finite difference based numerical scheme with utilization of Successive Over Relaxation (SOR) Method, Simpson’s (1/3) rule. Accuracy for this solution is assured through their computations for three grids sizes (h, h/2, h/4) and their closeness with the results of related published studies in the limiting case. In addition, extrapolation routine raises the order of accuracy to higher level. Table 4 indicates, how the results are rectified with reduction of step size and extrapolation.
Moreover, this work is attempted to develop better understanding of the micropolar structure of viscous fluids for the flow and thermal characteristics in porous medium over a permeable stretching sheet, we observed, heat transfer rate, shear and couple stresses at the sheet surface and the velocity, microrotation and temperature fields in the boundary layers over the sheet. It is noticed that Skin friction coefficient -f′′(0) and local Nusselt number -θ′(0) are lesser in value for micropolar fluids than for Newtonian fluids. Couple stress -N′(0) increases with increase in C₁( micropolar parameter).
Comment #6: The solution was implemented with iterative calculations. Therefore, it will be interesting to discuss the computational efficiency of the presented solutions.
Authors’ response: We are agreed with this comment. The computational code is developed and run in the environment of ForTran-90 on a simple desktop computer of Pantium 3 model. The iterative procedure worked very efficiently. The solution converged after small number of iterations. In compliance of the wise suggestion, Table 5 is now presented to show the enumeration of iteration with variation of step size and suction injection parameter. The selection of optimum value of relaxation parameter also played its role to increase efficiency of the iterative procedure.
Moreover:
- Authors would like to express their sincere appreciation to the expert reviewer who spared his precious time to critically evaluate the manuscript. His valuable suggestions have surely helped to improve the quality of this work.
- The manuscript has been converted in word format as advised by the editorial representative so a different style than that of Scientific work place format appears in the manuscript. Kindly accept the new style too as it will be easy for journal’s management body to handle.
- The changes are highlighted as red in the manuscript accordingly on the advises / suggestions of the 3 different reviewers and in two rounds of revisions.
- Kindly note it too that the paper under review with original title “Numerical Solution of Non-Linear Differential Equations Governing the Flow and Heat Transfer of Micropolar Fluids in a Porous Stretching Domain” was presented in World Congress ICIAM 2019 physically as oral talk by the undersigned (corresponding author of the paper) in July 2019, Valencia, Spain. Certificate of participation is attached/pasted below
- More over the journal “Mathematics” announced special issue tiled “Selected Papers from Iterative Processes for Solving Nonlinear Problems: Convergence and Stability of ICIAM 2019 and MME&HB 2019”, so the authors submitted the paper in the special issue meant for selected papers from ICIAM 2019.
- Kindly accept our explanations and allow us to carry on for the special issue tiled “Selected Papers from Iterative Processes for Solving Nonlinear Problems: Convergence and Stability of ICIAM 2019 and MME&HB 2019” of “Mathematics”. We have amended title of the article now on the advices of the other reviewer i.e. “Numerical Solution of Nonlinear Diff. Equations for Heat Transfer in Micropolar Fluids over a Stretching Domain”
- We are much obliged about the advises and comments of the reviewer and sure we are fully agreed with the comments of the reviewer.

Reviewer 2 Report
The corrections made as part of my first review have been addressed. The authors have taken into consideration all my questions and comments. I recommend to accept the new version of the paper.
Author Response
Round # 2: Authors’ Response to Comments of Reviewer 2
- Authors would like to express their sincere appreciation to the expert reviewer who spared his precious time to critically evaluate the manuscript. His valuable suggestions have surely helped to improve the quality of this work.
- The manuscript has been converted in word format as advised by the editorial representative so a different style than that of Scientific work place format appears in the manuscript. Kindly accept the new style too as it will be easy for journal’s management body to handle.
- The changes are highlighted as red in the manuscript accordingly on the advises / suggestions of the 3 different reviewers and in two rounds of revisions.
- Kindly note it too that the paper under review with original title “Numerical Solution of Non-Linear Differential Equations Governing the Flow and Heat Transfer of Micropolar Fluids in a Porous Stretching Domain” was presented in World Congress ICIAM 2019 physically as oral talk by the undersigned (corresponding author of the paper) in July 2019, Valencia, Spain. Certificate of participation is attached/pasted below
- More over the journal “Mathematics” announced special issue tiled “Selected Papers from Iterative Processes for Solving Nonlinear Problems: Convergence and Stability of ICIAM 2019 and MME&HB 2019”, so the authors submitted the paper in the special issue meant for selected papers from ICIAM 2019.
- Kindly accept our explanations and allow us to carry on for the special issue tiled “Selected Papers from Iterative Processes for Solving Nonlinear Problems: Convergence and Stability of ICIAM 2019 and MME&HB 2019” of “Mathematics”. We have amended title of the article now on the advices of the other reviewer i.e. “Numerical Solution of Nonlinear Diff. Equations for Heat Transfer in Micropolar Fluids over a Stretching Domain”
- We are much obliged about the advises and comments of the reviewer and sure we are fully agreed with the comments of the reviewer.

Reviewer 3 Report
Although the authors gave detailed answers to my comments, I cannot completely agree with their point of view. Undoubtedly, the paper may be of interest to a practical engineer, but the mathematical analysis of the problem under consideration is clearly insufficient for publication in a mathematical journal.
I believe that in its current form the paper is not suitable for publication in Mathematics. A major revision of the manuscript is required. The following comments will help authors make such revision.
(1) An analysis of the solvability of a nonlinear boundary value problem (2.12)--(2.15) and the qualitative properties of solutions is required. Why do the authors believe that this problem is well posed?
(2) In Introduction section, the authors give an extensive description of the literature, but this description is more formal than informative. Many sources are very distantly related to the problem under consideration, for example, [9], [10], [15], [16], [26] and others. However, there is no necessary comparison and clear description of what is the advantage and novelty of the proposed work.
(3) The following phrase should be clarified. See Page 4,
"The current work seems more practicable for thermal oil recovery and mega heat transfer requirements in large size industries as it involves generalized fluid (micropolar) instead of ordinary fluids (Newtonian)."
Author Response
Authors’ Responses
We are really thankful to the expert reviewer for sparing time to review the manuscript in detail and delivered value able suggestions. We agreed with these suggestions and had revised the manuscript accordingly.
- As per advice, we considered the solvability of the non-linear boundary value problem. A new section on “Existence and Uniqueness Results” is added. To best of our knowledge, it is first solvability attempt about this type of flow problem of micropolar fluids. It has added to the quality of manuscript. An effeort has been made to strengthen the mathematical analysis of the problem. The section on Mathematical Analysis is improved sufficiently. Moreover, the section on Numerical Method is also improved in Mathematical terms.
- The introduction section has been revised and updated. An integral part on inherited Non-linear nature of flow problems is introduced with reference of working solution procedures in the existing literature. We make our point clear that why we used finite difference based numerical scheme. It works well for simple geometries and simple grids. In this case its efficiency, accuracy and convergence are acceptable. Moreover, the sources distantly related to the problem as pointed out by the worthy reviewer are removed with agreement to the wise suggestion.
- The confusing phrase on page 4 is removed and the paragraph is rephrased.
- Authors would like to express their sincere appreciation to the expert reviewer who spared his precious time to critically evaluate the manuscript. His valuable suggestions have surely helped to improve the quality of this work.
- The manuscript has been converted in word format as advised by the editorial representative so a different style than that of Scientific work place format appears in the manuscript. Kindly accept the new style too as it will be easy for journal’s management body to handle.
- The changes are highlighted as red in the manuscript accordingly on the advises / suggestions of the 3 different reviewers and in two rounds of revisions.
- Kindly note it too that the paper under review with original title “Numerical Solution of Non-Linear Differential Equations Governing the Flow and Heat Transfer of Micropolar Fluids in a Porous Stretching Domain” was presented in World Congress ICIAM 2019 physically as oral talk by the undersigned (corresponding author of the paper) in July 2019, Valencia, Spain. Certificate of participation is attached/pasted below
- More over the journal “Mathematics” announced special issue tiled “Selected Papers from Iterative Processes for Solving Nonlinear Problems: Convergence and Stability of ICIAM 2019 and MME&HB 2019”, so the authors submitted the paper in the special issue meant for selected papers from ICIAM 2019.
- Kindly accept our explanations and allow us to carry on for the special issue tiled “Selected Papers from Iterative Processes for Solving Nonlinear Problems: Convergence and Stability of ICIAM 2019 and MME&HB 2019” of “Mathematics”. We have amended title of the article now on the advices of the other reviewer i.e. “Numerical Solution of Nonlinear Diff. Equations for Heat Transfer in Micropolar Fluids over a Stretching Domain”
We are much obliged about the advises and comments of the reviewer and sure we are fully agreed with the comments of the reviewer.

Round 3
Reviewer 3 Report
Following my comments, the authors made a major revision of the paper and the quality of the work undoubtedly improved. However, new flaws arose in the manuscript. These flaws need to be addressed.
(i) The proof of Lemmas 1 and 2, as well as the Theorem on Page 8, lacks the necessary details. Therefore, it is impossible to judge whether these statements are true or not.
(ii) On Page 7 the authors give the heading for section "Existence and Uniqueness Results". This is a bit surprising. In reality, the authors obtain only the existence results, but they do not give the uniqueness theorem. In such situation, the heading "Existence Results" looks more suitable.
Conclusion
I believe that after the introduction of appropriate corrections, the paper can be recommended for publication.
Author Response
Authors’ Response
We are grateful to the expert reviewer for his critical comments to improve the quality of the manuscript.
We are much agreed with his suggestions and guidelines. We have done our best to improve quality and to follow all suggestion of the honorable reviewer.
It is submitted that
- The proof of Lemmas 1 and 2, as well as the Theorem of existence of solution are presented in the manuscript and highlighted as Red.
- The section "Existence and Uniqueness Results” is now replaced with "Existence Results" in the manuscript.
Really, we got a lot of knowledge and skills from the guidelines and suggestions of the honorable reviewer, these will also improve our skills of writing a good paper in the future.
Thanks once again.

Round 4
Reviewer 3 Report
The authors have done a good work in implementing my review comments.
I believe that now the manuscript can be published.